# Verifiable Reinforcement Learning
# via Policy Extraction

**Osbert Bastani**
MIT
obastani@csail.mit.edu

**Yewen Pu**
MIT
yewenpu@mit.edu

**Armando Solar-Lezama**
MIT
asolar@csail.mit.edu

## Abstract

While deep reinforcement learning has successfully solved many challenging control tasks, its real-world applicability has been limited by the inability to ensure the safety of learned policies. We propose an approach to verifiable reinforcement learning by training decision tree policies, which can represent complex policies (since they are nonparametric), yet can be efficiently verified using existing techniques (since they are highly structured). The challenge is that decision tree policies are difficult to train. We propose VIPER, an algorithm that combines ideas from model compression and imitation learning to learn decision tree policies guided by a DNN policy (called the *oracle*) and its $Q$-function, and show that it substantially outperforms two baselines. We use VIPER to (i) learn a provably robust decision tree policy for a variant of Atari Pong with a symbolic state space, (ii) learn a decision tree policy for a toy game based on Pong that provably never loses, and (iii) learn a provably stable decision tree policy for cart-pole. In each case, the decision tree policy achieves performance equal to that of the original DNN policy.

## 1 Introduction

Deep reinforcement learning has proven to be a promising approach for automatically learning policies for control problems [11, 22, 29]. However, an important challenge limiting real-world applicability is the difficulty ensuring the safety of deep neural network (DNN) policies learned using reinforcement learning. For example, self-driving cars must robustly handle a variety of human behaviors [26], controllers for robotics typically need stability guarantees [2, 20, 8], and air traffic control should provably satisfy safety properties including robustness [19]. Due to the complexity of DNNs, verifying these properties is typically very inefficient if not infeasible [6].

Our goal is to learn policies for which desirable properties such as safety, stability, and robustness can be efficiently verified. We focus on learning decision tree policies for two reasons: (i) they are nonparametric, so in principle they can represent very complex policies, and (ii) they are highly structured, making them easy to verify. However, decision trees are challenging to learn even in the supervised setting; there has been some work learning decision tree policies for reinforcement learning [13], but we find that they do not even scale to simple problems like cart-pole [5].

To learn decision tree policies, we build on the idea of *model compression* [10] (or *distillation* [17]), which uses high-performing DNNs to guide the training of shallower [4, 17] or more structured [34, 7] classifiers. Their key insight is that DNNs perform better not because they have better representative power, but because they are better regularized and therefore easier to train [4]. Our goal is to devise a *policy extraction* algorithm that distills a high-performing DNN policy into a decision tree policy.

Our approach to policy extraction is based on *imitation learning* [27, 1], in particular, DAGGER [25]—the pretrained DNN policy (which we call the *oracle*) is used to generate labeled data, and then supervised learning is used to train a decision tree policy. However, we find that DAGGER learns

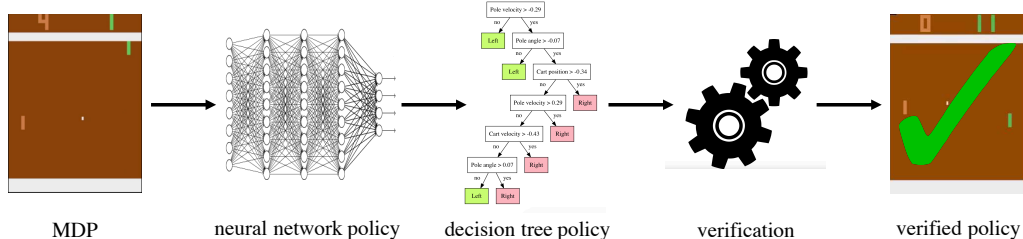

Figure 1: The high level approach VIPER uses to learn verifiable policies.

much larger decision tree policies than necessary. In particular, DAGGER cannot leverage the fact that our oracle provides not just the optimal action to take in a given state, but also the cumulative reward of every state-action pair (either directly as a $Q$-function or indirectly as a distribution over possible actions). First, we propose $Q$-DAGGER, a novel imitation learning algorithm that extends DAGGER to use the $Q$-function for the oracle; we show that $Q$-DAGGER can use this extra information to achieve provably better performance than DAGGER. Then, we propose VIPER[1], which modifies $Q$-DAGGER to extract decision tree policies; we show that VIPER can learn decision tree policies that are an order of magnitude smaller than those learned by DAGGER (and are thus easier to verify).

We show how existing verification techniques can be adapted to efficiently verify desirable properties of extracted decision tree policies: (i) we learn a decision tree policy that plays Atari Pong (on a symbolic abstraction of the state space rather than from pixels[2]) [22] and verify its robustness [6, 19], (ii) we learn a decision tree policy to play a toy game based on Pong, and prove that it never loses (the difficulty doing so for Atari Pong is that the system dynamics are unavailable)[3] and (iii) we learn a decision tree policy for cart-pole [5], and compute its region of stability around the goal state (with respect to the degree-5 Taylor approximation of the system dynamics). In each case, our decision tree policy also achieves perfect reward. Additionally, we discover a counterexample to the correctness of our decision tree policy for the toy game of pong, which we show can be fixed by slightly extending the paddle length. In summary, our contributions are:

- We propose an approach to learning verifiable policies (summarized in Figure 1).

- We propose a novel imitation learning algorithm called VIPER, which is based on DAGGER but leverages a $Q$-function for the oracle. We show that VIPER learns relatively small decision trees ($< 1000$ nodes) that play perfectly on Atari Pong (with symbolic state space), a toy game based on Pong, and cart-pole.

- We describe how to verify correctness (for the case of a toy game based on Pong), stability, and robustness of decision tree policies, and show that verification is orders of magnitude more scalable than approaches compatible with DNN policies.

**Related work.** There has been work on verifying machine learning systems [3, 30, 16, 6, 19, 18, 15]. Specific to reinforcement learning, there has been substantial interest in safe exploration [23, 36, 33]; see [14] for a survey. Verification of learned controllers [24, 32, 3, 20, 19, 31] is a crucial component of many such systems [2, 8], but existing approaches do not scale to high dimensional state spaces.

There has been work training decision tree policies for reinforcement learning [13], but we find that their approach does not even scale to cart-pole. There has also been work using model compression to learn decision trees [34, 7], but the focus has been on supervised learning rather than reinforcement learning, and on interpretability rather than verification. There has also been recent work using program synthesis to devise structured policies using imitation learning [35], but their focus is interpretability, and they are outperformed by DNNs even on cart-pole.

Figure 2: An MDP with initial state $s_0$, deterministic transitions shown as arrows (the label is the action), actions $A = \{\text{left, right, down}\}$ (taking an unavailable action transitions to $s_{\text{end}}$), rewards $R(\tilde{s}) = T$, $R(s_k) = T - \alpha$ (where $\alpha \in (0, 1)$ is a constant), and $R(s) = 0$ otherwise, and time horizon $T = 3(k + 1)$. Trajectories taken by $\pi^*$, $\pi_{\text{left}} : s \mapsto \text{left}$, and $\pi_{\text{right}} : s \mapsto \text{right}$ are shown as dashed edges, red edges, and green edges, respectively.

## 2 Policy Extraction

We describe $Q$-DAGGER, a general policy extraction algorithm with theoretical guarantees improving on DAGGER's, and then describe how VIPER modifies $Q$-DAGGER to extract decision tree policies.

**Problem formulation.** Let $(S, A, P, R)$ be a finite-horizon ($T$-step) MDP with states $S$, actions $A$, transition probabilities $P : S \times A \times S \to [0, 1]$ (i.e., $P(s, a, s') = p(s' \mid s, a)$), and rewards $R : S \to \mathbb{R}$. Given a policy $\pi : S \to A$, for $t \in \{0, ..., T - 1\}$, let

$$V_t^{(\pi)}(s) = R(s) + \sum_{s' \in S} P(s, \pi(s), s') V_{t+1}^{(\pi)}(s')$$

$$Q_t^{(\pi)}(s, a) = R(s) + \sum_{s' \in S} P(s, a, s') V_{t+1}^{(\pi)}(s')$$

be its value function and $Q$-function for $t \in \{0, ..., T - 1\}$, where $V_T^{(\pi)}(s) = 0$. Without loss of generality, we assume that there is a single initial state $s_0 \in S$. Then, let

$$d_0^{(\pi)}(s) = \mathbb{I}[s = s_0]$$

$$d_t^{(\pi)}(s) = \sum_{s' \in S} P(s', \pi(s'), s) d_{t-1}^{(\pi)}(s') \quad \text{(for } t > 0\text{)}$$

be the distribution over states at time $t$, where $\mathbb{I}$ is the indicator function, and let $d^{(\pi)}(s) = T^{-1} \sum_{t=0}^{T-1} d_t^{(\pi)}(s)$. Let $J(\pi) = -V_0^{(\pi)}(s_0)$ be the cost-to-go of $\pi$ from $s_0$. Our goal is to learn the best policy in a given class $\Pi$, leveraging an *oracle* $\pi^* : S \to A$ and its $Q$-function $Q_t^{(\pi^*)}(s, a)$.

**The $Q$-DAGGER algorithm.** Consider the (in general nonconvex) loss function

$$\ell_t(s, \pi) = V_t^{(\pi^*)}(s) - Q_t^{(\pi^*)}(s, \pi(s)).$$

Let $g(s, \pi) = \mathbb{I}[\pi(s) \neq \pi^*(s)]$ be the 0-1 loss and $\tilde{g}(s, \pi)$ a convex upper bound (in the parameters of $\pi$), e.g., the hinge loss [25].[4] Then, $\tilde{\ell}_t(s, \pi) = \tilde{\ell}_t(s)\tilde{g}(s, \pi)$ convex upper bounds $\ell_t(s, \pi)$, where

$$\tilde{\ell}_t(s) = V_t^{(\pi^*)}(s) - \min_{a \in A} Q_t^{(\pi^*)}(s, a).$$

$Q$-DAGGER runs DAGGER (Algorithm 3.1 from [25]) with the convex loss $\tilde{\ell}_t(s, \pi)$ and $\beta_i = \mathbb{I}[i = 1]$.

**Theory.** We bound the performance of $Q$-DAGGER and compare it to the bound in [25]; proofs are in Appendix A. First, we characterize the loss $\ell(\pi) = T^{-1} \sum_{t=0}^{T-1} \mathbb{E}_{s \sim d_t^{(\pi)}}[\ell_t(s, \pi)]$.

**Algorithm 1** Decision tree policy extraction.

---
**procedure** VIPER$((S, A, P, R), \pi^*, Q^*, M, N)$
    Initialize dataset $\mathcal{D} \leftarrow \varnothing$
    Initialize policy $\hat{\pi}_0 \leftarrow \pi^*$
    **for** $i = 1$ **to** $N$ **do**
        Sample $M$ trajectories $\mathcal{D}_i \leftarrow \{(s, \pi^*(s)) \sim d^{(\hat{\pi}_{i-1})}\}$
        Aggregate dataset $\mathcal{D} \leftarrow \mathcal{D} \cup \mathcal{D}_i$
        Resample dataset $\mathcal{D}' \leftarrow \{(s, a) \sim p((s, a)) \propto \tilde{\ell}(s)\mathbb{I}[(s, a) \in \mathcal{D}]\}$
        Train decision tree $\hat{\pi}_i \leftarrow \text{TrainDecisionTree}(\mathcal{D}')$
    **end for**
    **return** Best policy $\hat{\pi} \in \{\hat{\pi}_1, ..., \hat{\pi}_N\}$ on cross validation
**end procedure**

---

**Lemma 2.1.** *For any policy $\pi$, we have $T\ell(\pi) = J(\pi) - J(\pi^*)$.*

Next, let $\varepsilon_N = \min_{\pi \in \Pi} N^{-1} \sum_{i=1}^{N} T^{-1} \sum_{t=0}^{T-1} \mathbb{E}_{s \sim d_t^{(\hat{\pi}_i)}}[\tilde{\ell}_t(s, \pi)]$ be the training loss, where $N$ is the number of iterations of $Q$-DAGGER and $\hat{\pi}_i$ is the policy computed on iteration $i$. Let $\ell_{\max}$ be an upper bound on $\tilde{\ell}_t(s, \pi)$, i.e., $\tilde{\ell}_t(s, \pi) \leq \ell_{\max}$ for all $s \in S$ and $\pi \in \Pi$.

**Theorem 2.2.** *For any $\delta > 0$, there exists a policy $\hat{\pi} \in \{\hat{\pi}_1, ..., \hat{\pi}_N\}$ such that*

$$J(\hat{\pi}) \leq J(\pi^*) + T\varepsilon_N + \tilde{O}(1)$$

*with probability at least $1 - \delta$, as long as $N = \tilde{\Theta}(\ell_{max}^2 T^2 \log(1/\delta))$.*

In contrast, the bound $J(\hat{\pi}) \leq J(\pi^*) + uT\varepsilon_N + \tilde{O}(1)$ in [25] includes the value $u$ that upper bounds $Q_t^{(\pi^*)}(s, a) - Q_t^{(\pi^*)}(s, \pi^*(s))$ for all $a \in A$, $s \in S$, and $t \in \{0, ..., T-1\}$. In general, $u$ may be $O(T)$, e.g., if there are *critical states* $s$ such that failing to take the action $\pi^*(s)$ in $s$ results in forfeiting all subsequent rewards. For example, in cart-pole [5], we may consider the system to have failed if the pole hit the ground; in this case, all future reward is forfeited, so $u = O(T)$.

An analog of $u$ appears implicitly in $\varepsilon_N$, since our loss $\tilde{\ell}_t(s, \pi)$ includes an extra multiplicative factor $\tilde{\ell}_t(s) = V_t^{(\pi^*)}(s) - \min_{a \in A} Q_t^{(\pi^*)}(s, a)$. However, our bound is $O(T)$ as long as $\hat{\pi}$ achieves high accuracy on critical states, whereas the bound in [25] is $O(T^2)$ regardless of how well $\hat{\pi}$ performs.

We make the gap explicit. Consider the MDP in Figure 2 (with $\alpha \in (0, 1)$ constant and $T = 3(k+1)$). Let $\Pi = \{\pi_{\text{left}} : s \mapsto \text{left}, \pi_{\text{right}} : s \mapsto \text{right}\}$, and let $g(\pi) = \mathbb{E}_{s \sim d^{(\pi)}}[g(s, \pi)]$ be the 0-1 loss.

**Theorem 2.3.** *$g(\pi_{left}) = O(T^{-1})$, $g(\pi_{right}) = O(1)$, $\ell(\pi_{left}) = O(1)$, and $\ell(\pi_{right}) = O(T^{-1})$.*

That is, according to the 0-1 loss $g(\pi)$, the worse policy $\pi_{\text{left}}$ ($J(\pi_{\text{left}}) = 0$) is better, whereas according to our loss $\ell(\pi)$, the better policy $\pi_{\text{right}}$ ($J(\pi_{\text{right}}) = -(T - \alpha)$) is better.

**Extracting decision tree policies.** Our algorithm VIPER for extracting decision tree policies is shown in Algorithm 1. Because the loss function for decision trees is not convex, there do not exist online learning algorithms with the theoretical guarantees required by DAGGER. Nevertheless, we use a heuristic based on the follow-the-leader algorithm [25]—on each iteration, we use the CART algorithm [9] to train a decision tree on the aggregated dataset $\mathcal{D}$. We also assume that $\pi^*$ and $Q^{(\pi^*)}$ are not time-varying, which is typically true in practice. Next, rather than modify the loss optimized by CART, it resamples points $(s, a) \in \mathcal{D}$ weighted by $\tilde{\ell}(s)$, i.e., according to

$$p((s, a)) \propto \tilde{\ell}(s)\mathbb{I}[(s, a) \in \mathcal{D}].$$

Then, we have $\mathbb{E}_{(s,a) \sim p((s,a))}[\tilde{g}(s, \pi)] = \mathbb{E}_{(s,a) \sim \mathcal{D}}[\tilde{\ell}(s, \pi)]$, so using CART to train a decision tree on $\mathcal{D}'$ is in expectation equivalent to training a decision tree with the loss $\tilde{\ell}(s, \pi)$. Finally, when using neural network policies trained using policy gradients (so no $Q$-function is available), we use the maximum entropy formulation of reinforcement learning to obtain $Q$ values, i.e., $Q(s, a) = \log \pi^*(s, a)$, where $\pi^*(s, a)$ is the probability that the (stochastic) oracle takes action $a$ in state $s$ [37].

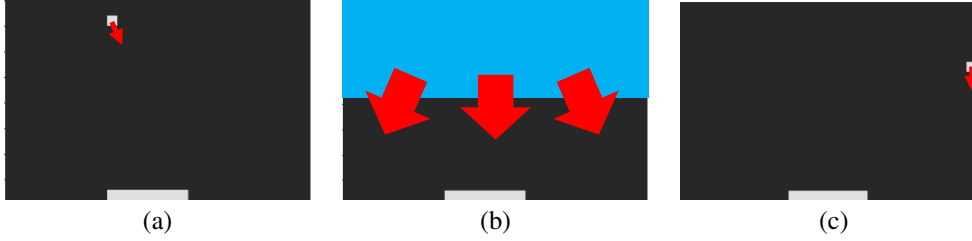

|  (a) | (b) | (c) |

Figure 3: (a) An example of an initial state of our toy pong model; the ball is the white dot, the paddle is the white rectangle at the bottom, and the red arrow denotes the initial velocity $(v_x, v_y)$ of the ball. (b) An intuitive visualization of the ball positions (blue region) and velocities (red arrows) in $Y_0$. (c) A counterexample to correctness discovered by our verification algorithm.

## 3   Verification

In this section, we describe three desirable control properties we can efficiently verify for decision tree policies but are difficult to verify for DNN policies.

**Correctness for toy Pong.**   Correctness of a controller is system-dependent; we first discuss proving correctness of controller for a toy model of the Pong Atari game [22]. This toy model consists of a ball bouncing on the screen, with a player-controlled paddle at the bottom. If the ball hits the top or the side of the screen, or if the ball hits the paddle at the bottom, then it is reflected; if the ball hits the bottom of the screen where the paddle is not present, then the game is over. The system is frictionless and all collisions are elastic. It can be thought of as Pong where the system paddle is replaced with a wall. The goal is to play for as long as possible before the game ends. The states are $(x, y, v_x, v_y, x_p) \in \mathbb{R}^5$, where $(x, y)$ is the position of the ball (with $x \in [0, x_{\max}]$ and $y \in [0, y_{\max}]$), $(v_x, v_y)$ is its velocity (with $v_x, v_y \in [-v_{\max}, v_{\max}]$), and $x_p$ is the position of the paddle (with $x_p \in [0, x_{\max}]$), and the actions are $\{\text{left}, \text{right}, \text{stay}\}$, indicating how to move the paddle.

Our goal is to prove that the controller never loses, i.e., the ball never hits the bottom of the screen at a position where the paddle is not present. More precisely, assuming the system is initialized to a safe state (i.e., $y \in Y_0 = [y_{\max}/2, y_{\max}]$), then it should avoid an unsafe region (i.e., $y = 0 \wedge (x \le x_p - L \vee x \ge x_p + L)$, where $L$ is half the paddle length).

To do so, we assume that the speed of the ball in the $y$ direction is lower bounded, i.e., $|v_y| > v_{\min}$; since velocity in each direction is conserved, this assumption is equivalent to assuming that the initial $y$ velocity is in $[-v_{\max}, -v_{\min}] \cup [v_{\min}, v_{\max}]$. Then, it suffices to prove the following inductive invariant: as long as the ball starts in $Y_0$, then it re-enters $Y_0$ after at most $t_{\max} = \lceil 2y_{\max}/v_{\min} \rceil$ steps.

Both the dynamics $f : S \times A \to S$ and the controller $\pi : S \to A$ are piecewise-linear, so the joint dynamics $f_\pi(s) = f(s, \pi(s))$ are also piecewise linear; let $S = S_1 \cup ... \cup S_k$ be a partition of the state space so that $f_\pi(s) = f_i(s) = \beta_i^T s$ for all $s \in S_i$. Then, let $s_t$ be a variable denoting the state of the system at time $t \in \{0, ..., t_{\max}\}$; then, the following constraints specify the system dynamics:

$$\phi_t = \bigvee_{i=1}^{k} (s_{t-1} \in S_i \Rightarrow s_t = \beta_i^T s_{t-1}) \quad \forall t \in \{1, ..., t_{\max}\}$$

Furthermore letting $\psi_t = (s_t \in Y_0)$, we can express the correctness of the system as the formula[5]

$$\psi = \left( \bigwedge_{t=1}^{t_{\max}} \phi_t \right) \wedge \psi_0 \Rightarrow \bigvee_{t=1}^{t_{\max}} \psi_t.$$

Note that $\sigma \Rightarrow \tau$ is equivalent to $\neg \sigma \vee \tau$. Then, since $Y_0$ and all of the $S_i$ are polyhedron, the predicates $s_t \in Y_0$ and $s_t \in S_i$ are conjunctions of linear (in)equalities; thus, the formulas $\psi_t$ and $\phi_t$ are disjunctions of conjunctions of linear (in)equalities. As a consequence, $\psi$ consists of conjunctions and disjunctions of linear (in)equalities; standard tools exist for checking whether such formulas

are satisfiable [12]. In particular, the controller is correct if and only if $\neg\psi$ is unsatisfiable, since a satisfying assignment to $\neg\psi$ is a counterexample showing that $\psi$ does not always hold.

Finally, note that we can slightly simplify $\psi$: (i) we only have to show that the system enters a state where $v_y > 0$ after $t_{\max}$ steps, not that it returns to $Y_0$, and (ii) we can restrict $Y_0$ to states where $v_y < 0$. We use parameters $(x_{\max}, y_{\max}, v_{\min}, v_{\max}, L) = (30, 20, 1, 2, 4)$; Figure 3 (a) shows an example of an initial state, and Figure 3 (b) depicts the set $Y_0$ of initial states that we verify.

**Correctness for cart-pole.**   We also discuss proving correctness of a cart-pole control policy. The classical cart-pole control problem has a 4-dimensional state space $(x, v, \theta, \omega) \in \mathbb{R}^4$, where $x$ is the cart position, $v$ is the cart velocity, $\theta$ is the pole angle, and $\omega$ is the pole angular velocity, and a 1-dimensional action space $a \in \mathbb{R}$, where $a$ is the lateral force to apply to the cart. Consider a controller trained to move the cart to the right while keeping the pole in the upright position. The goal is to prove that the pole never falls below a certain height, which can be encoded as the formula[6]

$$\psi \equiv s_0 \in S_0 \land \bigwedge_{t=0}^{\infty} |\phi(s_t)| \leq y_0,$$

where $S_0 = [-0.05, 0.05]^4$ is the set of initial states, $s_t = f(s_{t-1}, a_{t-1})$ is the state on step $t$, $f$ is the transition function, $\phi(s)$ is the deviation of the pole angle from upright in state $s$, and $y_0$ is the maximum desirable deviation from the upright position. As with correctness for toy Pong, the controller is correct if $\neg\psi$ is unsatisfiable. The property $\psi$ can be thought of as a toy example of a safety property we would like to verify for a controller for a walking robot—in particular, we might want the robot to run as fast as possible, but prove that it never falls over while doing so. There are two difficulties verifying $\psi$: (i) the infinite time horizon, and (ii) the nonlinear transitions $f$. To address (i), we approximate the system using a finite time horizon $T_{\max} = 10$, i.e., we show that the system is safe for the first ten time steps. To address (ii), we use a linear approximation $f(s, a) \approx As + Ba$; for cart-pole, this approximation is good as long as $\phi(s_t)$ is small.

**Stability.**   Stability is a property from control theory saying that systems asymptotically reach their goal [31]. Consider a continuous-time dynamical system with states $s \in S = \mathbb{R}^n$, actions $a \in A = \mathbb{R}^m$, and dynamics $\dot{s} = f(s, a)$. For a policy $\pi : S \to A$, we say the system $f_\pi(s) = f(s, \pi(s))$ is *stable* if there is a *region of attraction* $U \subseteq \mathbb{R}^n$ containing 0 such that for any $s_0 \in U$, we have $\lim_{t \to \infty} s(t) = 0$, where $s(t)$ is a solution to $\dot{s} = f(s, a)$ with initial condition $s(0) = s_0$.

When $f_\pi$ is nonlinear, we can verify stability (and compute $U$) by finding a *Lyapunov function* $V : S \to \mathbb{R}$ which satisfies (i) $V(s) > 0$ for all $s \in U \setminus \{0\}$, (ii) $V(0) = 0$, and (iii) $\dot{V}(s) = (\nabla V)(s) \cdot f(s) < 0$ for all $s \in U \setminus \{0\}$ [31]. Given a *candidate* Lyapunov function, exhaustive search can be used to check whether the Lyapunov properties hold [8], but scales exponentially in $n$.

When $f_\pi$ is polynomial, we can use sum-of-squares (SOS) optimization to devise a candidate Lyapunov function, check the Lyapunov properites, and compute $U$ [24, 32, 31]; we give a brief overview. First, suppose that $V(s) = s^T P s$ for some $P \in \mathbb{R}^{n \times n}$. To compuate a candidate Lyapunov function, we choose $P$ so that the Lyapunov properties hold for the linear approximation $f_\pi(s) \approx As$, which can be accomplished by solving the SOS program [7]

$$\exists P \in \mathbb{R}^{n \times n} \tag{1}$$
$$\text{subj. to} \quad s^T P s - \|s\|^2 \geq 0 \text{ and } s^T P A s + \|s\|^2 \leq 0 \quad (\forall s \in S).$$

The first equation ensures properties (i) and (ii)—in particular, the term $\|s\|^2$ ensures that $s^T P s > 0$ except when $s = 0$. Similarly, the second equation ensures property (iii). Next, we can simultaneously check whether the Lyapunov properties hold for $f_\pi$ and compute $U$ using the SOS program

$$\arg\max_{\rho \in \mathbb{R}_+, \Lambda \in \mathbb{R}^{n \times n}} \rho \tag{2}$$
$$\text{subj. to} \quad (s^T \Lambda s)(s^T P f_\pi(s)) + (\rho - s^T P s)\|s\|^2 \leq 0 \text{ and } s^T \Lambda s \geq 0 \quad (\forall s \in S).$$

The term $\lambda(s) = s^T \Lambda s$ is a slack variable—when $\rho > s^T P s$ or $s = 0$ (so the second term is nonpositive), it can be made sufficiently large so that the first constraint holds regardless of $s^T P f_\pi(s)$,

but when $\rho \leq s^T P s$ and $s \neq 0$ (so the second term is positive), we must have $s^T P f_\pi(s) < 0$ since $s^T \Lambda s \geq 0$ by the second constraint. Properites (i) and (ii) hold from (1), and (2) verifies (iii) for all

$$s \in U = \{ s \in S \mid V(s) \leq \rho \}.$$

Thus, if a solution $\rho > 0$ is found, then $V$ is a Lyapunov function with region of attraction $U$. This approach extends to higher-order polynomials $V(s)$ by taking $V(s) = m(s)^T P m(s)$, where $m(s)$ is a vector of monomials (and similarly for $\lambda(s)$).

Now, let $\pi$ be a decision tree whose leaf nodes are associated with linear functions of the state $s$ (rather than restricted to constant functions). For $\ell \in \mathrm{leaves}(\pi)$, let $\beta_\ell^T s$ be the associated linear function. Let $\ell_0 \in \mathrm{leaves}(\pi)$ be the leaf node such that $0 \in \mathrm{routed}(\ell_0, \pi)$, where $\mathrm{routed}(\ell; \pi) \subseteq S$ is the set of states routed to $\ell$ (i.e., the computation of the decision tree maps $s$ to leaf node $\ell$). Then, we can compute a Lyapunov function for the linear policy $\tilde{\pi}(s) = \beta_{\ell_0}^T s$; letting $\tilde{U}$ be the region of attraction for $\tilde{\pi}$, the region of attraction for $\pi$ is $U = \tilde{U} \cap \mathrm{routed}(\ell_0, \pi)$. To maximize $U$, we can bias the decision tree learning algorithm to prefer branching farther from $s = 0$.

There are two limitations of our approach. First, we require that the dynamics be polynomial. For convenience, we use Taylor approximations of the dynamics, which approximates the true property but works well in practice [32]. This limitation can be addressed by reformulating the dynamics as a polynomial system or by handling approximation error in the dynamics [31]. Second, we focus on verifying stability locally around 0; there has been work extending the approach we use by "patching together" different regions of attraction [32].

**Robustness.** Robustness has been studied for image classification [30, 16, 6]. We study this property primarily since it can be checked when the dynamics are unknown, though it has been studied for air traffic control as a safety consideration [19]. We say $\pi$ is $\varepsilon$-*robust* at $s_0 \in S = \mathbb{R}^d$ if[8]

$$\pi(s) = \pi(s_0) \quad (\forall s \in B_\infty(s_0, \varepsilon)),$$

where $B_\infty(s_0, \varepsilon)$ is the $L_\infty$-ball of radius $\varepsilon$ around $s_0$. If $\pi$ is a decision tree, we can efficiently compute the largest $\varepsilon$ such that $\pi$ is $\varepsilon$-robust at $s_0$, which we denote $\varepsilon(s_0; \pi)$. Consider a leaf node $\ell \in \mathrm{leaves}(\pi)$ labeled with action $a_\ell \neq \pi(s_0)$. The following linear program computes the distance from $s_0$ to the closest point $s \in S$ (in $L_\infty$ norm) such that $s \in \mathrm{routed}(\ell; \pi)$:

$$\varepsilon(s_0; \ell, \pi) = \max_{s \in S, \varepsilon \in \mathbb{R}_+} \varepsilon$$

$$\text{subj. to } \left( \bigwedge_{n \in \mathrm{path}(\ell; \pi)} \delta_n s_{i_n} \leq t_n \right) \wedge \left( \bigwedge_{i \in [d]} |s_i - (s_0)_i| \leq \varepsilon \right),$$

where $\mathrm{path}(\ell; \pi)$ is the set of internal nodes along the path from the root of $\pi$ to $\ell$, $\delta_n = 1$ if $n$ is a left-child and $-1$ otherwise, $i_n$ is the feature index of $n$, and $t_n$ is the threshold of $n$. Then,

$$\varepsilon(s_0; \pi) = \arg \min_{\ell \in \mathrm{leaves}(\pi)} \begin{cases} \infty & \text{if } a_\ell = \pi(s_0) \\ \varepsilon(s_0; \pi, \ell) & \text{otherwise.} \end{cases}$$

## 4 Evaluation

**Verifying robustness of an Atari Pong controller.** For the Atari Pong environment, we use a 7-dimensional state space (extracted from raw images), which includes the position $(x, y)$ and velocity $(v_x, v_y)$ of the ball, and the position $y_p$, velocity $v_p$, acceleration $a_p$, and jerk $j_p$ of the player's paddle. The actions are $A = \{\mathrm{up}, \mathrm{down}, \mathrm{stay}\}$, corresponding to moving the paddle up, down, or unchanged. A reward of 1 is given if the player scores, and -1 if the opponent scores, for 21 rounds (so $R \in \{-21, ..., 21\}$). Our oracle is the deep $Q$-network [22], which achieves a perfect reward of 21.0 (averaged over 50 rollouts). [9] VIPER (with $N = 80$ iterations and $M = 10$ sampled traces per iteration) extracts a decision tree policy $\pi$ with 769 nodes that also achieves perfect reward 21.0.

We compute the robustness $\varepsilon(s_0; \pi)$ at 5 random states $s_0 \in S$, which took just under 2.9 seconds for each point (on a 2.5 GHz Intel Core i7 CPU); the computed $\varepsilon$ varies from 0.5 to 2.8. We compare to

Reluplex, a state-of-the-art tool for verifying DNNs. We use policy gradients to train a stochastic DNN policy $\pi : \mathbb{R}^7 \times A \to [0, 1]$, and use Reluplex to compute the robustness of $\pi$ on the same 5 points. We use line search on $\varepsilon$ to find the distance to the nearest adversarial example to within 0.1 (which requires 4 iterations of Reluplex); in contrast, our approach computes $\varepsilon$ to within $10^{-5}$, and can easily be made more precise. The Reluplex running times varied substantially—they were 12, 136, 641, and 649 seconds; verifying the fifth point timed out after running for one hour.

**Verifying correctness of a toy Pong controller.** Because we do not have a model of the system dynamics for Atari Pong, we cannot verify correctness; we instead verify correctness for our toy model of Pong. We use policy gradients to train a DNN policy to play toy pong, which achieves a perfect reward of 250 (averaged over 50 rollouts), which is the maximum number of time steps. VIPER extracts a decision tree with 31 nodes, which also plays perfectly. We use Z3 to check satisfiability of $\neg\psi$. In fact, we discover a counterexample—when the ball starts near the edge of the screen, the paddle oscillates and may miss it.[10]

Furthermore, by manually examining this counterexample, we were able to devise two fixes to repair the system. First, we discovered a region of the state space where the decision tree was taking a clearly suboptimal action that led to the counterexample. To fix this issue, we added a top-level node to the decision tree so that it performs a safer action in this case. Second, we noticed that extending the paddle length by one (i.e., $L = 9/2$) was also sufficient to remove the counterexample. For both fixes, we reran the verification algorithm and proved that the no additional counterexamples exist, i.e., the controller never loses the game. All verification tasks ran in just under 5 seconds.

**Verifying correctness of a cart-pole controller.** We restricted to discrete actions $a \in A = \{-1, 1\}$, and used policy gradients to train a stochastic oracle $\pi^* : S \times A \to [0, 1]$ (a neural network with a single hidden layer) to keep the pole upright while moving the cart to the right; the oracle achieved a perfect reward of 200.0 (averaged over 100 rollouts), i.e., the pole never falls down. We use VIPER as before to extract a decision tree policy. In Figure 4 (a), we show the reward achieved by extracted decision trees of varying sizes—a tree with just 3 nodes (one internal and two leaf) suffices to achieve perfect reward. We used Z3 to check satisfiability of $\neg\psi$; Z3 proves that the desired safety property holds, running in 1.5 seconds.

**Verifying stability of a cart-pole controller.** Next, we tried to verify stability of the cart-pole controller, trained as before except without moving the cart to the right; as before, the decision tree achieves a perfect reward of 200.0. However, achieving a perfect reward only requires that the pole does not fall below a given height, not stability; thus, neither the extracted decision tree policy nor the original neural network policy are stable.

Instead, we used an approach inspired by guided policy search [21]. We trained another decision tree using a different oracle, namely, an iterative linear quadratic regulator (iLQR), which comes with stability guarantees (at least with respect to the linear approximation of the dynamics, which are a very good near the origin). Note that we require a model to use an iLQR oracle, but we anyway need the true model to verify stability. We use iLQR with a time horizon of $T = 50$ steps and $n = 3$ iterations. To extract a policy, we use $Q(s, a) = -J_T(s)$, where $J_T(s) = s^T P_T s$ is the cost-to-go for the final iLQR step. Because iLQR can be slow, we compute the LQR controller for the linear approximation of the dynamics around the origin, and use it when $\|s\|_\infty \leq 0.05$. We now use continuous actions $A = [-a_{\max}, a_{\max}]$, so we extract a (3 node) decision tree policy $\pi$ with linear regressors at the leaves (internal branches are axis-aligned); $\pi$ achieves a reward of 200.0.

We verify stability of $\pi$ with respect to the degree-5 Taylor approximation of the cart-pole dynamics. Solving the SOS program (2) takes 3.9 seconds. The optimal solution is $\rho = 3.75$, which suffices to verify that the region of stability contains $\{s \in S \mid \|s\|_\infty \leq 0.03\}$. We compare to an enumerative algorithm for verifying stability similar to the one used in [8]; after running for more than 10 minutes, it only verified a region $U'$ whose volume is $10^{-15}$ that of $U$. To the best of our knowledge, enumeration is the only approach that can be used to verify stability of neural network policies.

**Comparison to fitted $Q$ iteration.** On the cart-pole benchmark, we compare VIPER to fitted $Q$ iteration [13], which is an actor-critic algorithm that uses a decision tree policy that is retrained on

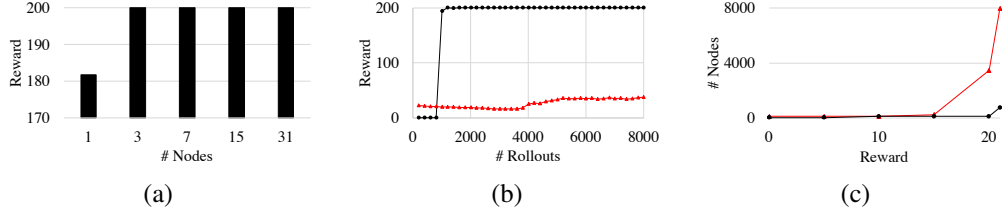

Figure 4: (a) Reward (maximum $R = 200$) as a function of the size (in number of nodes) of the decision tree extracted by VIPER, on the cart-pole benchmark. (b) Reward (maximum $R = 200$) as a function of the number of training rollouts, on the cart-pole benchmark, for VIPER (black, circle) and fitted $Q$-iteration (red, triangle); for VIPER, we include rollouts used to train the oracle. (c) Decision tree size needed to achieve a given reward $R \in \{0, 5, 10, 15, 20, 21\}$ (maximum $R = 21$), on the Atari Pong benchmark, for VIPER (black, circle) and DAGGER with the 0-1 loss (red, triangle).

every step rather than using gradient updates; for the $Q$-function, we use a neural network with a single hidden layer. In Figure 4 (b), we compare the reward achieved by VIPER compared to fitted $Q$ iteration as a function of the number of rollouts (for VIPER, we include the initial rollouts used to train the oracle $\pi^*$). Even after 200K rollouts, fitted $Q$ iteration only achieves a reward of 104.3.

**Comparison to** DAGGER. On the Atari Pong benchmark, we compare VIPER to using DAGGER with the 0-1 loss. We use each algorithm to learn decision trees with maximum depths from 4 to 16. In Figure 4 (c), we show the smallest size decision tree needed to achieve reward $R \in \{0, 5, 10, 15, 20, 21\}$. VIPER consistently produces trees an order of magnitude smaller than those produced by DAGGER—e.g., for $R = 0$ (31 nodes vs. 127 nodes), $R = 20$ (127 nodes vs. 3459 nodes), and $R = 21$ (769 nodes vs. 7967 nodes)—likely because VIPER prioritizes accuracy on critical states. Evaluating pointwise robustness for DAGGER trees is thus an order of magnitude slower: 36 to 40 seconds for the $R = 21$ tree (vs. under 3 seconds for the $R = 21$ VIPER tree).

**Controller for half-cheetah.** We demonstrate that we can learn high quality decision trees for the half-cheetah problem instance in the MuJoCo benchmark. In particular, we used a neural network oracle trained using PPO [28] to extract a regression tree controller. The regression tree had 9757 nodes, and achieved cumulative reward $R = 4014$ (whereas the neural network achieved $R = 4189$).

## 5 Conclusion

We have proposed an approach to learning decision tree policies that can be verified efficiently. Much work remains to be done to fully realize the potential of our approach. For instance, we used a number of approximations to verify correctness for the cart-pole controller; it may be possible to avoid these approximations, e.g., by finding an invariant set (similar to our approach to verifying toy Pong), and by using upper and lower piecewise linear bounds on transition function. More generally, we considered a limited variety of verification tasks; we expect that a wider range of properties may be verified for our policies. Another important direction is exploring whether we can automatically repair errors discovered in a decision tree policy. Finally, our decision tree policies may be useful for improving the efficiency of safe reinforcement learning algorithms that rely on verification.

### Acknowledgments

This work was funded by the Toyota Research Institute and NSF InTrans award 1665282.

## Footnotes

[1] VIPER stands for Verifiability via Iterative Policy ExtRaction.

[2] We believe that this limitation is reasonable for safety-critical systems; furthermore, a model of the system dynamics defined with respect to symbolic state space is anyway required for most verification tasks.

[3] We believe that having the system dynamics are available is a reasonable assumption; they are available for most real-world robots, including sophisticated robots such as the walking robot ATLAS [20].

[4] Other choices of $\tilde{g}$ are possible; our theory holds as long as it is a convex upper bound on the 0-1 loss $g$.

[5]We are verifying correctness over a continuous state space, so enumerative approaches are not feasible.

[6]This property cannot be expressed as a stability property since the cart is always moving.

[7]Simpler approaches exist, but this one motivates our approach to checking whether the Lyapunov properties hold for $V$ for the polynomial dynamics $f_\pi$.

[8]This definition of robustness is different than the one in control theory.

[9]This policy operates on images, but we can still use it as an oracle.

[10]While this counterexample was not present for the original neural network controller, we have no way of knowing if other counterexamples exist for that controller.

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
