[Reviews · NeurIPS 2018]

Reviewer 1



Post rebuttal Thank the authors for the clarification. One minor point I realised is the equation between line 144 and 145. Is this constraint really a disjunction over partitions? If there is at least one partition the given state doesn't belong to, it would be always true because at least one of inner propositions will be true, wouldn't it? ------------------------------------------------ This paper proposed a policy extraction algorithm, Viper, which learns a decision tree policy imitating the given oracle (e.g. DNN policy). The trained decision tree policy allows for its verification in terms of, more specifically, correctness, stability and robustness. Viper imitates the given oracle policy using its internal sub-algorithm, Q-Dagger, which extends Dagger (Ross et al., 2011). Q-Dagger addresses two limitations of Dagger, such as 1) its tendency to output too large decision tree than necessary and 2) its lack of consideration of actions’ long-term effects (e.g. via Q-value). The first point is empirically proven in the cart-pole benchmark (Figure 4(c)) showing Viper results in much smaller decision trees than Dagger at the same level of reward. The second is achieved by a new loss function based on negative advantage w.r.t. the given policy (\tilde l_t(s,\pi)), and resampling dataset in proportion to its extreme (\tilde l_t(s)) if this reviewer is not mistaken. The benefits of decision tree policy are explained in Section 3, and demonstrated in the experiment section, on Pong and Cart-Pole environments. Quality The paper is well written and appears technically sound, but some technical details are not clear (see Clarity section) Clarity One issue with Q-Dagger is that the proof for Theorem 2.2 doesn’t look clear enough as it simply relies on the proof of Theorem 4.2 in (Ross et al. 2011). Although Q-Dagger is a variant of Dagger with the new loss function and a fixed $\beta$, the readers who are not very familiar with this might get confused easily. What’s the benefit of the hinge loss in regret? Also, it would be great to discuss how the heuristics introduced to Viper impact this theoretical result. Minor concerns - Is this theoretical result applicable to continuous state space? - Line 353: In the second line of the equation, $V^{(\pi)}_{t+1}$ -> $V^{(\pi*)}_{t+1}$. - Citation for Reluplex is missing. Originality and significance Overall it looks a nice contribution. A new algorithm for extracting decision tree policies is proposed with tighter bound, and its effectiveness in verification is demonstrated.

Reviewer 2



# Paper ID 1246 Verifiable RL via Policy Extraction ## Summary The main idea of the paper is to use a strong learned policy as an expert (e.g., deep neural network) in an imitation learning setup. A loss function designed to minimize the gap between the best and worst action (as defined by the expert) is used to train a CART policy. Three verifiable properties (correctness, robustness, stability) can be associated with the tree from the piecewise linearity of the partitions imposed by the tree policy that the original expert policy (e.g., a deep neural network) may not possess. However, correctness and stability require the system dynamics to be known. The empirical evaluation is somewhat limited. For example, experiments are shown on two relatively small domains (Atari Pong, cart-pole) where the model dynamics are simple enough to specify or approximate. Overall, the paper is clearly well written. The algorithmic ideas are intuitively clear and seem sound to me. However, I'm a bit unclear about the feasibility / applicability of the verification process in other domains. The experimental section is also a bit limited. Overall, there are some interesting ideas in the paper but it's a bit difficult for me to evaluate due to the limited discussion and experiments. Detailed comments follow. ## Detailed Comments - The paper introduces a vanilla imitation learning setup for the task of extracting a policy from a given oracle or expert. - A loss function designed to minimize the worst-case gap across the expected state distributions is defined. A decision tree learner is used on a dataset resampled under the new loss function which has the effect of "compiling" the original expert into a decision tree trained to minimize the expected worst-case loss across the distribution of states induced by the learned tree policy. This learning algorithm (Q-DAGGER) is the key algorithmic contribution. - Intuitively, the main idea is to focus the training on states where the gap in the values between the best action and the worst action is largest. These are the "critical states". Standard 0-1 loss doesn't treat these states differently from other states. The idea here is that the learner should focus on avoiding the worst-case loss in states (i.e., $l_max$ is small). If such a policy can be found, then the bound improves over vanilla DAGGER. - It's not clear to me how problems with "critical states" are distributed. For example, are the common control tasks in MuJoCo / OpenGym good candidate problems for this approach? Additional discussion characterizing problems where the methods are most / least applicable would be nice to have. - Assuming such a problem is given, the approach proceeds by modifying the loss function to incorporate the worst-case gap in values. This requires an expert that can provide a worst-case "gap" (e.g., Q values, expert policy probabilities) between the best and worst action in any state. This seems reasonable to me. - A tree policy can now be learned via CART in an iterative fashion (i.e. DAGGER). The main idea is to reweight the accumulated examples in the dataset using the worst-case gap values described above. This is the policy learned by the VIPER algorithm. This part is also very straightforward and well described. - Next, the paper proposes three properties that are verifiable by VIPER under appropriate conditions. Correctness and stability require the model dynamics to be known while robustness does not. Here, I do think there needs to be additional discussion on the feasibility of obtaining the dynamics for other problems. Also, even if the dynamics are known, the difficulty of specifying the constraints in the form required by VIPER is unclear to me. A discussion of the complexity of these properties on MuJoCo tasks, for example, may be good to have. Discussion of these in Section 3 would strengthen the paper significantly, in my opinion. - The empirical evaluation is performed on the Pong domain and cart-pole. It seems like verifying correctness and stability is rather difficult for a given domain. It might be worth discussing. Next, I'd be curious to know why robustness is only evaluated for Atari Pong and not cart-pole? Perhaps the paper could focus on a single property (e.g., robustness) and show experiments on additional domains. In its current form, the experimental section feels a bit limited. Additional experiments and their analysis would be very nice to have. - Overall, it seems like the choice of loss function is orthogonal to the verification requirements, which only require the decision tree. Is my understanding correct? If yes, then other loss functions and their impact on tree size and properties would be quite interesting to study. For example, I'm curious to understand why exactly the trees under the 0-1 loss are larger for a particular reward target. How much of the tree size depends on the worst-case loss function seems like something worth exploring further. Update -------- I thank the authors for their response. My overall score remains unchanged.

Reviewer 3



This paper uses policy extraction/distillation to describe a complex policy in terms of a decision tree. Given the tree, the paper then can prove properties w.r.t. correctness (given a model of the environment), stability, and robustness. The idea of providing provably correct policies is very appealing for reasons that the paper describes well. However, it's not clear to me that this paper takes a critical step in this direction. In particular, the correctness property is most compelling, and it requires a full model of the environment (which typically will not be available in real-world domains). It's not clear to me w.r.t. the correctness counterexample found for the toy pong controller 1) If it was "obvious" that changing the paddle length fixed the counter example, 2) Why changing the paddle length was preferable to changing the policy, and 3) If it was possible that additional counterexamples existed. ----- Typos I found: Verifying correcntess which are a very good near the origin ---------------- Post-Rebuttal After reading the other reviews and the authors' rebuttal I have moved my score slightly up. I am still hesitant to recommend acceptance because I had trouble understanding some parts of the paper (section 3 was pretty far out of my area) and because it's not clear to me how well this will apply to real-world problems. If the other reviewers are confident in their understanding of the validation section, I would not be opposed to accepting, particularly since I believe this work is reasonably novel.